# The Cationic Amphiphilic Drug Hexamethylene Amiloride Eradicates Bulk Breast Cancer Cells and Therapy-Resistant Subpopulations with Similar Efficiencies

**DOI:** 10.3390/cancers14040949

**Published:** 2022-02-14

**Authors:** Anastasia L. Berg, Ashley Rowson-Hodel, Michelle Hu, Michael Keeling, Hao Wu, Kacey VanderVorst, Jenny J. Chen, Jason Hatakeyama, Joseph Jilek, Courtney A. Dreyer, Madelyn R. Wheeler, Ai-Ming Yu, Yuanpei Li, Kermit L. Carraway

**Affiliations:** 1Department of Biochemistry and Molecular Medicine, University of California, Sacramento, CA 95817, USA; alberg@ucdavis.edu (A.L.B.); arhodel@ucdavis.edu (A.R.-H.); mghu@ucdavis.edu (M.H.); mtkeeling@yahoo.com (M.K.); hhwu@ucdavis.edu (H.W.); kvandervorst@ucdavis.edu (K.V.); wxlchen@ucdavis.edu (J.J.C.); jhatake@ucdavis.edu (J.H.); jljilek@pharmacy.arizona.edu (J.J.); cadreyer@ucdavis.edu (C.A.D.); mrkring@ucdavis.edu (M.R.W.); aimyu@ucdavis.edu (A.-M.Y.); lypli@ucdavis.edu (Y.L.); 2Davis Comprehensive Cancer Center, University of California Davis School of Medicine, Sacramento, CA 95817, USA

**Keywords:** breast cancer, cancer stem cell, therapy resistance, cationic amphiphilic drug, lysosome-dependent cell death

## Abstract

**Simple Summary:**

A limitation to successful therapeutic outcomes for breast and other cancer patients is the ability of small subsets of tumor cells to resist the apoptotic cell death provoked by currently employed therapeutic agents. These therapy-resistant cancer stem cell populations can then seed recurrent tumors and metastatic lesions, compromising the efficacy of the treatment regimen. The aim of our study was to assess the hypothesis that cationic amphiphilic drugs (CADs), which induce tumor cell death via the unrelated programmed necrotic mechanism, exhibit efficacy toward cancer stem cell populations that are resistant to currently employed therapeutics. We found that the therapy-resistant stem-like subpopulation of cells from a variety of breast cancer models are as sensitive to CADs as the bulk population. Our observations imply that the incorporation of cationic amphiphilic anticancer agents into existing therapeutic regimens could ultimately improve breast cancer patient outcomes by minimizing tumor recurrence and metastatic outgrowth.

**Abstract:**

The resistance of cancer cell subpopulations, including cancer stem cell (CSC) populations, to apoptosis-inducing chemotherapeutic agents is a key barrier to improved outcomes for cancer patients. The cationic amphiphilic drug hexamethylene amiloride (HMA) has been previously demonstrated to efficiently kill bulk breast cancer cells independent of tumor subtype or species but acts poorly toward non-transformed cells derived from multiple tissues. Here, we demonstrate that HMA is similarly cytotoxic toward breast CSC-related subpopulations that are resistant to conventional chemotherapeutic agents, but poorly cytotoxic toward normal mammary stem cells. HMA inhibits the sphere-forming capacity of FACS-sorted human and mouse mammary CSC-related cells in vitro, specifically kills tumor but not normal mammary organoids ex vivo, and inhibits metastatic outgrowth in vivo, consistent with CSC suppression. Moreover, HMA inhibits viability and sphere formation by lung, colon, pancreatic, brain, liver, prostate, and bladder tumor cell lines, suggesting that its effects may be applicable to multiple malignancies. Our observations expose a key vulnerability intrinsic to cancer stem cells and point to novel strategies for the exploitation of cationic amphiphilic drugs in cancer treatment.

## 1. Introduction

Studies in both hematological and solid tumor malignancies support the notion that tumor-initiating or cancer stem cells (CSCs)—a rare, relatively quiescent, and highly tumorigenic cancer cell population endowed with the capacity for self-renewal, anchorage independence, and multilineage differentiation—harbor intrinsic therapy resistance mechanisms and pose significant clinical challenges [1,2,3,4,5]. CSCs are resistant to cellular stresses [6], persist during therapeutic intervention [7], convey substantial tumor chemoresistance [8], potentiate post-therapy recurrence [9], and are strongly associated with progressive, metastatic disease [7].

While a potentially promising strategy, therapeutically targeting CSCs presents unique obstacles considering their notorious resistance to apoptotic death and capacity to launch primary tumor recurrence and metastases, even in cases of apparent complete clinical remission [3,10]. Insensitivity to caspase-dependent apoptotic signaling is among the most important factors conferring enhanced tumor growth, survival, and resistance to traditional chemotherapeutics and targeted drugs [11,12,13]. Accordingly, novel therapeutics that engage non-apoptotic cell death pathways to kill CSCs and other apoptosis-resistant tumor cells warrant investigation.

We have reported that 5-(N,N-hexamethylene) amiloride (HMA), a derivative of the FDA-approved potassium-sparing diuretic amiloride and a member of the broader class of cationic amphiphilic drugs (CADs) [14,15,16], is cytotoxic toward cultured breast cancer cells independent of caspase activity [17]. HMA reduces the viability of breast cancer cells of differing molecular profiles with equal efficiency [17], which is significant as breast cancer subtypes (luminal or basal, estrogen/progesterone receptor-positive (ER/PR+), HER2−amplified (HER2+), or receptor-negative (ER/PR/HER2−)) are variably resistant to chemotherapeutics and targeted drugs [18]. Further, HMA is equally effective in eradicating both dividing and non-dividing cells [17], distinguishing it from cytotoxic agents that are employed clinically to kill actively proliferating cells but that can leave behind the quiescent CSC population responsible for tumor repopulation. Clinical disease recurrence, in fact, is often observed following use of commonly employed antimitotic agents (taxanes, vinca alkaloids, kinase inhibitors, etc.) [19].

Fundamental to HMA’s ability to deplete heterogeneous tumor cell populations is its capacity to induce a form of necrotic cell death initiated by disruption of lysosomal function and requiring lysosomal cathepsin activity for cell death [17], unlike commonly cited forms (e.g., necroptosis and parthanatos). HMA induces aggregation of acidic vesicles and formation of lysosomal multilamellar bodies, indicative of perturbed lysosomal membrane dynamics [17]. Indeed, lysosomes have been described as ‘suicide-bags’ based on the hydrolytic capacity of sequestered enzymes [20], and cell autolysis following lysosomal membrane permeabilization (LMP) has been shown to mediate cancer cell death [21,22,23,24]. Direct engagement of LMP leading to lysosome-dependent cell death (LDCD) has been demonstrated for a limited number of CADs and has been postulated as an attractive strategy for the eradication of apoptosis-resistant cancer cells [22,25]. Recent findings reveal that cancer cell lysosomes are particularly fragile, with heightened susceptibility to LMP [26]. Cells undergo characteristic morphological and functional changes during the process of malignant transformation, including alterations in lysosomal volume, quantity, membrane structure and composition, and hydrolase activity [20,24,26]. Paradoxically, these transformation-associated changes that promote tumor growth and invasiveness also render cancer cell lysosomes unstable and represent a cancer-specific vulnerability that might be exploited for therapeutic purposes. However, the sensitivity of the CSC population to CAD-induced LMP is currently unknown.

Here, we establish the susceptibility of chemotherapy-insensitive cancer cells and CSC-related cells isolated from an array of tumor types to HMA-induced LDCD, leading to the novel suggestion that CADs may exhibit benefit as components of maintenance therapy following primary treatment.

## 2. Materials and Methods

### 2.1. Cell Culture

MDA-MB-231, MCF7, SKBR3, T47D, 4T1, nMuMG, T98G, Du145, A549, HepG2, J82, LS174T, Panc-1, and SKOV3 cells were purchased from American Type Culture Collection (ATCC, Manassas, VA, USA) and maintained as recommended at 37° in 10% CO_2_ in media supplemented with 10% fetal bovine serum (FBS; Genesee Scientific, San Diego, CA, USA) and antibiotics (penicillin/streptomycin; Thermo Fisher, Waltham, MA, USA). MCF10A (ATCC) cells were grown in DMEM/F12 (#SH30023; HyClone, Logan, UT, USA) base medium supplemented with 5% horse serum (Thermo Fisher), 20 ng/mL EGF, 0.5 mg/mL hydrocortisone, 100 ng/mL cholera toxin, 10 μg/mL insulin (Millipore-Sigma, Burlington, MA, USA), and 1% penicillin/streptomycin (Thermo Fisher). HMEC4 cells were maintained in mammary epithelial basal media (MEBM; #CC-3151, Lonza, Basel, Switzerland) with MEGM SingleQuots Supplements (#CC-4136, Lonza). Met-1 (gifted by A.D. Borowsky, UC Davis School of Medicine, Sacramento, CA, USA) and NDL cells were maintained as previously described [27,28]. Cell lines were authenticated prior to use by short-tandem repeat profiling (Genetics Core Facility; University of Arizona, Tucson, AZ, USA) and were replaced with a cryopreserved stock every six passages. Mouse brain tissue was dissociated as described [29,30], and primary cells were cultured in DMEM base medium (#11995065, Thermo Fisher) for no more than one passage. Cell line attributes are summarized in Appendix A.

### 2.2. Cell Viability Assays

Trypan blue staining was carried out as described previously [17] and counted using a Countess™ II Automated Cell Counter (Thermo Fisher). For MTT assays, media from treated cells was replaced with 5 mg/mL 3-(4,5-Dimethyl-2-thiazolyl)-2,5-diphenyl-2H-tetrazolium bromide (MTT, #M5655; Millipore-Sigma) solution in base media for 1 h. Cells were washed with PBS, crystals were dissolved using isopropyl alcohol (IPA, 0.5% 1N HCl in isopropanol), and absorbance (λ_ex_ 570 nm) was measured with a FilterMax F5 microplate reader (Molecular Devices, San Jose, CA, USA) and Multi-Mode Analysis software (Version 3.4.0.27 Beckman Coulter, Brea, CA, USA).

### 2.3. Animal Therapeutic Studies

All experimental protocols were approved by the IACUC of the University of California, Davis, USA. The MMTV-NDL mouse model has previously been described [31]. Wild-type females were crossed with NDL males to generate WT and NDL mice, and genotypes were confirmed by polymerase chain reaction using primers for NDL (Fwd-TTCCGGAACCCACATCAG; Rev-GTTTCCTGCAGCAGCCTA).

Seven- to ten-week-old female FVB/NJ (#001800) mice were purchased from The Jackson Laboratory (Bar Harbor, ME, USA) and allowed to acclimate for at least one week before use. For evaluation of intratumoral drug delivery, 3 × 10^6^ syngeneic Met-1 cells in PBS were mixed in a 1:1 volume/volume mixture of PuraMatrix Peptide Hydrogel (#354250; Corning, Corning, NY, USA) and injected bilaterally into the fourth mammary fat pads of mice under anesthesia by continuous inhalation of 2% isoflurane gas. Sterile tweezers were used to lift the fourth nipple, and a syringe needle was used to implant cell suspensions directly into the mammary fat pad. Tumors were measured daily using digital calipers, and tumor volume was calculated according to the formula v = (w^2^ × l) × 0.5236. When tumors reached a volume of 100 mm^3^, animals were randomized into two cohorts (*n* = 3) and unilaterally administered 0.78 μg HMA per mm^3^ tumor in 10% DMSO in saline or DMSO–saline control daily for 7 days. Contralateral tumors were un-injected and served as internal controls. Animals were sacrificed 2 days following treatment end by CO_2_ asphyxiation; all tumors were collected and fixed in 10% neutral buffered formalin for paraffin embedding and sectioning, while lungs were fixed for whole-mount analysis.

For the experimental model of metastasis, 1 × 10^6^ NDL tumor cells in PBS were injected into the lateral tail vein of female FVB/NJ mice. After 7 days, randomized animals were treated intravenously with either HMA (30 mg/kg)-loaded disulfide cross-linked micelles (DCMs) or empty DCMs twice weekly for four weeks. Polymer synthesis, micelle preparation, drug loading, and micelle size and distribution characterization for in vivo delivery were carried out as previously described [32]. Animals were sacrificed 24 h after the final treatment, and tissues were harvested for analysis of metastasis. All mice were caged as mixed treatment cohorts.

### 2.4. Cell Labeling, Flow Cytometry, and Sorting

MDA-MB-231, MCF7, and SKBR3 human cancer cell lines were trypsinized to single cells, and MMTV-NDL murine mammary tumors (1.0–1.5 cm in diameter) were harvested and dissociated to single cells as previously described [29,30] with minor modifications. Cells were suspended at 1 × 10^7^ per mL in staining buffer (PBS with 2% FBS) and incubated for 30 min on ice with antibodies. Cells were washed three times, re-suspended in staining buffer with 1 μg/mL Propidium Iodide (PI), analyzed, and sorted with a FACS Aria II cell sorter (Becton Dickinson, Franklin Lakes, NJ, USA). Sorting schemes were based on previously published studies for human cell lines [1,8] and primary mouse tissues [33,34]. Results were analyzed using FlowJo software. Antibodies used were CD24-PE-Cy7 (1:100 dilution, #561646) and CD44-APC (1:100 dilution, #559942; BD Pharmingen, San Diego, CA, USA) for human cells, and CD24-PE (1:200 dilution, #553262; BD Pharmingen), CD49f-APC (1:100 dilution, #313615; Biolegend, San Diego, CA, USA), CD31-PE-Cy7 (1:100, #102417; Biolegend), and CD45-PE-Cy7 (1:100, #103113; Biolegend) for mouse cells.

### 2.5. Tumorsphere/Mammosphere Assays

For all sphere assays, single cells were plated at a density of 2–5 × 10^3^ cells per well on Corning Costar ultra-low attachment 24-well plates (#CLS3473, Sigma Aldrich), and spheres > 50 μm in diameter were quantified. FACS-sorted cells from human cell lines were plated in serum-free MammoCult base medium with proliferation supplement (#05620; Stem Cell Technologies, Vancouver, BC, Canada), and sorted cells from murine mammary tumors were plated in DMEM/F12 medium containing basic fibroblast growth factor (bFGF, 20 ng/mL, #354060, Corning), epidermal growth factor (EGF, 20 ng/mL, #354001, Corning), heparin (4 μg/mL, #07980, Stem Cell Technologies), and B-27 supplement (1:50 dilution, #17504044, Thermo Fisher). Cells were treated with vehicle (DMSO), HMA (#A9561, Sigma Aldrich), or chemotherapeutics (Cisplatin (#479306, Sigma Aldrich), Docetaxel (#S1148; Selleckchem, Houston, TX, USA), Doxorubicin (#S1208; Selleckchem)) prior to suspension.

For secondary sphere forming assays, cell lines were cultured to 70% confluency in two dimensions and treated with vehicle or HMA for 24 h. Cells were then trypsinized to single cells and plated in ultra-low attachment plates as follows: MDA-MB-231, MCF7, SKBR3, and T47D in MammoCult medium; MCF10A, HMEC4, and nMuMG in MEGM; T98G, Du145, A549, HepG2, J82, LS174T, Panc-1, and SKOV3 in 3D Tumorsphere Medium XF (#C-28070; PromoCell, Heidelberg, Germany). After 7 days, primary spheres were dissociated to single cells by trypsinization and re-plated in ultra-low attachment plates to grow for another 7 days. Secondary spheres were quantified.

For tumorsphere assays for chemoresistance, cell lines were cultured in two dimensions and treated with chemotherapeutics as indicated for 72 h. Single-cell suspensions were plated at equivalent densities in ultra-low attachment plates in serum-free MammoCult medium. A secondary dose of chemotherapeutic, HMA, or vehicle was then administered to cells in suspension, and tumorspheres were allowed to form over 7 days and then quantified. All images were captured using an Olympus IX81 inverted microscope and CellSens Entry software version 1.7.

### 2.6. Persister Cell Generation and Treatments

An MCF7 persister cell population was generated based on previous studies [35,36]. Parental MCF7 cells were seeded in 12-well plates and cultured to 70–80% confluency prior to treatment with 4 μM DTX. Cells were washed twice with PBS, and media with new drug were replaced every three days over a nine-day incubation period. MCF7 parental cells from the same persister line origin population were simultaneously cultured and plated on day 9 of persister line culture in 12-well plates, and both parental and persister cells were treated the next day with either vehicle control (DMSO), 4 μM DTX, or 40 μM HMA for 24 h. Cell death was evaluated by trypan blue assay.

### 2.7. Ex Vivo Organoid Assay for Drug Cytotoxicity

MMTV-PyMT and MMTV-NDL murine mammary tumors (1.0–1.5 cm in diameter) or pooled mammary glands from FVB-NJ mice were harvested and dissociated to single cells as previously described [29,30] with minor modifications. A total of 2 × 10^5^ single cells were embedded in Matrigel (#354230, Corning) in organoid growth medium on 24-well plates, which has been previously described [37]. Organoids were formed after 7 days in culture and were then exposed to vehicle and varying concentrations of HMA for 72 h. MMTV-PyMT and MMTV-NDL organoid viability was measured using RealTime Glo (#G9711; Promega, Madison, WI, USA) according to the manufacturer’s instructions. Images were taken before (day 0) drug treatment and daily over the course of drug treatment. Representative brightfield images were taken with an Olympus IX81 microscope with CellSens Entry software version 1.7; chemiluminescent images were taken with a ChemiDoc MP Imaging System (BioRad, Hercules, CA, USA) and analyzed with Image Lab software version 1.2 to quantify the RealTime Glo signal.

### 2.8. Establishment of Orthotopic Xenograft Mouse Models

All animal studies were performed in accordance with protocols approved by the Institutional Animal Care and Use Committee of the University of California, Davis. Eight- to twelve-week-old Balb/cJ (#000651) and FVB/NJ (#001800) mice were purchased from The Jackson Laboratory and allowed to acclimate for at least one week before use. Orthotopic mammary fat pad implantation was performed as follows: A total of 1–2 × 10^6^ syngeneic tumor cells (4T1 cells into Balb/cJ, Met-1 cells into FVB/NJ) suspended in PBS were mixed in a 1:1 volume/volume mixture of PuraMatrix Peptide Hydrogel (#354250; Corning) and injected into the fourth mammary fat pads of mice under anesthesia by continuous inhalation of 2% isoflurane gas. Sterile tweezers were used to lift the fourth nipple, and a syringe needle was used to implant cell suspensions directly into the mammary fat pad. Tumors were measured twice weekly using digital calipers, and mice were sacrificed when tumors reached 1.0–1.5 cm in diameter.

### 2.9. Organotypic Tumor Slice Preparation and Viability Analysis

Cores 4 mm in diameter were punched from sacrificed murine tumors using a biopsy punch (#7424; RoyalTek, TWN) and cut into 1 mm organotypic tumor slices. Tumor slices were cultured individually on 12 mm Transwell with 0.4 μm pore polycarbonate membrane inserts (#3401; Corning) using 12-well plates. Slices were cultured in 1× Advanced Dulbecco’s Modified Eagle Medium (#12491023; Thermo Fisher) supplemented with 5% FBS, 1× GlutaMAX (#35050061; Thermo Fisher), 0.5× Penicillin-Streptomycin, 1× Insulin-Transferrin-Selenium supplement (#41400045; Thermo Fisher), and 15 mM HEPES (#15630130; Thermo Fisher) as previously described [38] and maintained at 37° in 10% CO_2_. After 24 h in culture, slices were exposed to vehicle or 40 μM HMA for 24 h. Tumor slice viability was measured using RealTime Glo (#G9711; Promega) according to the manufacturer’s instructions. Images were taken before (day 0) drug treatment and 24 h after drug treatment with a ChemiDoc MP Imaging System (BioRad) and analyzed with Image Lab software version 1.2 to quantify the RealTime Glo signal.

### 2.10. Lung Analysis

Lungs were inflated with PBS, fixed in formalin for 24 h at 4 °C, and stained in carmine alum as described [39,40]. Gross lesions were imaged under a dissecting scope (Zeiss Stemi 2000-C; Axiocam ERc/5 s) and processed for confirmation by histology.

### 2.11. Histology

All drug-injected and un-injected contralateral control xenograft tumors were subjected to histological analysis. H&E-stained sections were prepared using previously described methods [39] and analyzed for the presence of necrosis in a blinded fashion. Area of active necrosis was quantified and normalized to total tumor area for 3–5 randomly selected fields across 3 serial sections. Immunohistochemistry was performed as previously described [41]. An internal negative control (no primary antibody) was included with each analysis.

### 2.12. Statistical, Data, and Image Analysis

Values are expressed as averages and were calculated from a minimum of three replicate experiments, unless otherwise stated. Statistical significance was established using a Student *t*-test or the Wilcoxon rank-sum test, with *p*-values less than 0.05 considered statistically significant. Data analysis and graphical representation were performed with Microsoft Excel or the R statistical platform. Images were compiled in Microsoft PowerPoint, with brightness and contrast altered only for presentation clarity.

## 3. Results

### 3.1. HMA Ablates Chemotherapy-Resistant BCSCs but Not Normal Mammary Stem-like Cells

Our previous studies demonstrated that HMA kills human- and mouse-derived mammary carcinoma cells, but not non-transformed cells from a variety of tissues, independent of proliferative state or molecular subtype [17]. These observations raise the possibility that HMA may be cytotoxic toward the relatively quiescent CSC population, and that HMA treatment may circumvent the chemotherapeutic resistance characteristic of breast cancer subtypes [18]. For example, receptor-negative breast carcinomas are enriched for a CSC phenotype [42] and display increased recurrence and chemoresistance rates relative to luminal breast cancers [18], yet we observed that analogous cultured cell models (MDA-MB-231 and MCF7) respond similarly to HMA [17].

To assess the ability of HMA to specifically eradicate breast CSCs (BCSCs), we first examined the impact of HMA pretreatment on secondary sphere formation [43,44,45] by cultured human breast cancer cell lines. We observed that the EC_50_s for HMA-mediated tumorsphere suppression toward bulk populations of MDA-MB-231 (ER/PR/HER2-), MCF7 and T47D (ER/PR+), and SKBR3 (HER2+) human breast cancer cell lines (see Appendix A) are essentially identical to each other (Figure 1A) and identical to the EC_50_s we previously reported for HMA cytotoxicity toward the total populations of these cell lines [17]. Likewise, CD44^+^/CD24^low^-sorted (see Appendix A) MDA-MB-231, MCF7, and SKBR3 BCSC-enriched subpopulations are similarly sensitive to HMA (Figure 1B). In contrast, EC_50_ levels of HMA did not reduce secondary sphere formation by the non-tumorigenic mammary epithelial human breast MCF10A and HMEC and mouse mammary nMuMG cell lines (Figure 1C). These observations confirm the hypothesis that HMA is similarly cytotoxic toward BCSC subpopulations and bulk populations and point to the existence of a therapeutic window that could circumvent a substantial barrier to CSC-targeted therapeutic development [46].

To discern whether therapy-resistant BCSC subpopulations are sensitive to HMA, we examined the sphere-forming capabilities of sorted cells after treatment with HMA or the apoptosis-inducing conventional chemotherapeutic agent cisplatin (CIS), or docetaxel (DTX) together with doxorubicin (DOX), at levels similar to their EC_50_s for cytotoxicity toward the total cell population (Figure 1D). We observed that while the CD44^+^/CD24^low^ subpopulations are resistant to these chemotherapeutic agents, they remain sensitive to HMA (Figure 1E and Appendix A). Given the inherent chemoresistance of BCSCs and their propensity to expand under selective pressures imparted by chemotherapy drugs [8], it was unsurprising that both CIS and combination DTX/DOX treatment exhibited enhanced sphere formation relative to the vehicle control.

We observed similar effects of HMA on the sphere outgrowth of BCSCs derived from mammary tumors arising in the MMTV-NDL transgenic mouse model of HER2-positive breast cancer. These mice develop metastatic multifocal adenocarcinomas at approximately 20 weeks of age [47,48]. The BCSC subpopulation was isolated by sorting (Appendix A) for markers CD24 and CD49f [33] and lack of lineage-specific markers (Lin^−^/CD31^−^/CD45^−^) [29,33,34]. The HMA EC_50_ for inhibition of mouse mammary tumorsphere outgrowth by sorted cells was comparable to that observed for human breast cancer cell lines (Figure 1F), and HMA dramatically reduced sphere growth when compared to chemotherapeutics CIS and DTX/DOX (Figure 1G and Appendix A).

### 3.2. HMA Is Cytotoxic toward Tumor Cell Populations Insensitive to Conventional Chemotherapeutics

To directly assess HMA’s ability to ameliorate chemotherapy resistance, we analyzed the chemotherapy-insensitive ‘persister’ cell population that survives acute cytotoxic assault [49]. Persister cells derived from the parental MCF7 line were generated through continuous 9-day culture in the presence of high-dose DTX (Appendix A) and displayed insensitivity toward further DTX administration, but exhibited sensitivity to HMA to a degree comparable to that observed in the MCF7 parental cell line (Figure 2A). Similarly, Met-1 mouse mammary carcinoma cells treated with high-dose CIS (Appendix A) or DTX (Appendix A) for 48 h demonstrated therapy insensitivity upon secondary CIS or DTX treatment but were sensitive to HMA. Furthermore, when DTX-resistant cells were plated in a tumorsphere assay following secondary DTX, HMA, or vehicle control administration, a significant reduction in sphere outgrowth from HMA-treated cells was observed, while the sphere count was equivalent between secondary DTX- and control-treated cells (Appendix A).

We further interrogated the capacity of HMA to target therapy-resistant human BCSCs by selecting for the BCSC subset in sphere-forming conditions following extended dosing with chemotherapy drugs. MDA-MB-231, MCF7, SKBR3, and T47D cell lines were cultured in the presence of DTX (Figure 2B), CIS (Figure 2C), or combination DTX/DOX (Figure 2D) for 72 h and subsequently plated in a tumorsphere assay. Secondary treatment with HMA yielded a dramatic reduction in sphere formation, while treatment with chemotherapies consistently enriched for a BCSC phenotype. These observations affirm the potential of HMA to overcome both single- and multi-drug resistance uniformly across a variety of breast cancer subtypes.

### 3.3. HMA Thwarts the Viability of Mouse Mammary Tumor Tissues Ex Vivo

To evaluate HMA’s efficacy in more physiologically relevant systems recapitulating the complexities of the in vivo environment, we adapted an ex vivo primary tumor organoid model [37] to incorporate fundamental properties of tissue architecture using the MMTV-NDL and MMTV-PyMT transgenic mouse models of breast cancer [47,50]. Fresh tumor tissues were dissociated to single cells and embedded in a three-dimensional Matrigel scaffold for 7-day culture to produce tumor organoids. Organoid viability over the course of subsequent 72 h HMA treatment was monitored in real time with the use of a non-lytic, bioluminescent method. We observed that HMA dramatically reduces both MMTV-PyMT (Figure 3A) and MMTV-NDL (Figure 3B) organoid viability over time, which can be distinguished by a dramatic change in organoid morphology and loss of structural integrity by brightfield imaging (Figure 3C,D). In contrast to the marked reduction in tumor-derived organoid viability observed in response to HMA (Figure 3D, left panels), the viability and morphology of organoids generated from matched normal FVB/NJ mouse mammary glands were unaffected by HMA treatment over 48 h (Figure 3D, right panels).

Recently, a novel organotypic tumor slice culture method was developed to interrogate the tumor immune microenvironment and cancer cell response to cytotoxic agents in the context of a heterogeneous tissue retaining a complex tumor architecture [51]. Cores are punched from fresh tumor tissues and then sliced for individual ex vivo culture on transwell inserts. We employed this technique to assess HMA’s impact on several mouse mammary tumor models, including the MMTV-NDL and MMTV-PyMT transgenic mouse models as well as two orthotopic syngeneic mammary xenograft models of aggressive disease (BALB/cJ and FVB/NJ mice transplanted with 4T1 and Met-1 mouse mammary tumor cells, respectively). Slice viability was verified by bioluminescence prior to drug administration and evaluated after 24 h of HMA treatment, revealing a striking loss of the viability signal (Figure 3E) and marked tumor cytotoxicity across all four models (Figure 3F).

### 3.4. HMA Induces Necrosis In Vivo and Suppresses Metastasis

We previously demonstrated that HMA induces the programmed necrotic death of cultured cancer cells by a lysosome-dependent mechanism [17]. To affirm that HMA invokes a comparable mode of cytotoxicity in vivo, we generated a syngeneic xenograft model of mammary tumorigenesis via transplantation of Met-1 mouse mammary carcinoma cells directly into the mammary glands of FVB/NJ mice. We observed that daily direct intratumoral injection of HMA for 7 days resulted in a significant increase in necrotic tumor area by histology relative to vehicle control-treated tumors as well as un-injected contralateral control tumors (Figure 4A,B). The modest necrosis observed in the vehicle control tumors may be accounted for by injection force shear. Met-1 cells are highly aggressive and readily metastasize to the lung, and notably, gross morphological and histological analysis of lung tissues collected from HMA-treated animals revealed fewer metastatic lesions than those from vehicle control-treated animals (Figure 4C,D). These findings are consistent with a reduced capacity for BCSCs to survive and establish metastatic foci following necrotic death induced by HMA.

We next sought to investigate whether HMA impacts metastatic colonization of tumor cells when injected directly into the vasculature. We established an experimental model of metastasis whereby NDL mouse mammary tumor cells were instilled into the lateral tail vein of FVB/NJ mice [39]. Systemic delivery of the naked HMA compound has little impact on overall tumor burden due to its short half-life in vivo [52,53]. To enhance drug bioavailability for these studies, we encapsulated HMA in reversibly disulfide cross-linked micelles (DCMs) that minimize premature drug release in the circulation. HMA-loaded (5 mg/mL) DCM nanoparticles were determined to have a uniform size by dynamic light scattering methods [32], indicating sufficiency for in vivo delivery (Appendix A), and exhibited equivalent cytotoxicity to the free drug (Appendix A). Beginning 7 days after cell instillation, animals were treated intravenously with 30 mg/kg HMA/DCM (based on studies of the maximum tolerated dose in FVB/NJ animals) or empty DCM control twice weekly for four weeks, and harvested lung tissues were analyzed by gross morphology and histology (Figure 4E). We observed a significant reduction in metastatic lung colonization with systemic HMA/DCM treatment relative to the DCM control (Figure 4F). Animal body weights were stable during systemic treatment (Appendix A), and no signs of gross organ damage were observed, indicating a lack of drug toxicity. These observations affirm that HMA effectively inhibits tumor metastasis and secondary tumor initiation at distant tissue sites in animal models of breast cancer, again consistent with its ability to eradicate the BCSC subpopulation.

### 3.5. HMA Depletes Cancer Cells Derived from an Array of Human Tissue Types and Inhibits Outgrowth of Enriched Cancer Stem-like Populations

Considering the remarkable consistency of HMA’s cytotoxic effects in diverse breast cancer cell lines and tissues as described here and in our previous report [17], we investigated whether HMA might be utilized as a pan-therapeutic for malignancies arising from other tissues. To address this question, we tested the impact of HMA on transformed human cell lines derived from an array of solid tissue tumor types (Appendix A). T98G, Du145, A549, HepG2, J82, LS174T, Panc-1, and SKOV3 cell lines all demonstrated a dose-dependent response to HMA (Figure 5A) and exhibited EC_50_ values very similar to bulk breast cancer cell lines and BCSCs. Notably, HMA is cytotoxic toward T98G cells derived from glioblastoma, an aggressive form of brain cancer with a poor clinical prognosis due to a distinct lack of therapeutic options [54,55], but not toward cells dissociated from fresh mouse brain tissue (Figure 5B).

We further examined HMA’s effects on non-breast CSCs using the tumorsphere assay. Low-adherent growth conditions enrich for the CSC population across cell types [56,57], and all cell lines tested displayed a statistically significant reduction in secondary sphere outgrowth following 24 h EC_50_ HMA treatment, with the exception of LS174T and Du145, which fell outside of significance because of the inherent variability in sphere formation across multiple biological replicates (Figure 5C,D). The highly aggressive and therapy-resistant lines T98G [58], Panc-1 [59], A549 [60], and HEPG2 [61] exhibited a marked reduction in secondary sphere formation in response to HMA compared to the control. Notably, we observed elevated levels of cellular debris in the HMA-treated samples, which, in combination with the reduced sphere number, indicated increased cellular stress and CSC death. The striking CSC morbidity reflected in the impaired sphere outgrowth observed across cell lines representing diverse solid tumor types underscores the potentially broad therapeutic applications of HMA’s mechanism of action in cancer treatment.

## 4. Discussion

Collectively, our findings reveal the broad efficacy of HMA-induced LDCD and underscore the growing promise of lysosome-targeting CADs and therapeutic strategies in cancer treatment. While lysosome-destabilizing compounds were evaluated as anticancer agents as early as the 1970s, the approach was temporarily abandoned due to a lack of a specific methodology available to identify LMP as the causative event in cell death [20,62,63]. Meanwhile, the evolving study of lysosomal protease biology revealed diverse activities for cathepsins in cell death and disease. Pro-tumorigenic roles for cathepsins (promotion of cancer cell proliferation, invasion, and metastasis through extracellular matrix remodeling and stimulation of angiogenesis) have been established [26,64,65,66]; conversely, cathepsins function in LMP-mediated cell death mechanisms in cancer and other developmental and disease contexts [22,67,68]. The elevated cathepsin expression observed in malignant transformation, together with an expansion of the lysosomal compartment and alterations in membrane protein composition, serves to enhance tumor aggressiveness. However, these characteristic features of cancer cell lysosomes augment cellular susceptibility to LMP, providing strong rationale for lysosome-based therapeutic intervention [69].

Indeed, various agents that induce LMP or otherwise affect lysosomal function are under investigation for cancer, with few in clinical development, though clinical efficacy data are presently lacking for these strategies [69,70]. Preclinical evaluation of the CAD siramesine revealed its ability to induce LMP and tumor-selective cell death [21] hinging on displacement of the lipid-metabolizing enzyme acid sphingomyelinase (ASM) from its lipid cofactor bis(monoacylglycero)phosphate (BMP), thereby inhibiting hydrolysis of sphingomyelin to ceramide and disrupting membrane dynamics due to sphingomyelin accumulation [24]. Transformation-associated loss of ASM expression purportedly accounts for the cancer sensitization observed. Accordingly, it will be prudent to identify HMA’s lysosomal molecular target and uncover whether a similar mechanism underlies its tumor cell selectivity.

Moreover, current FDA-approved drugs with CAD-like properties including antihistamines [71] and antimalarials [72] display antitumor effects in preclinical as well as epidemiological studies. For example, a retrospective analysis of Danish patients diagnosed with ovarian cancer concluded that prior use of CAD antihistamines was associated with a significant survival benefit [73]. Further considering the apparent safe long-term use of these clinical CADs—which exhibit lysosomotropic qualities including rapid organelle sequestration and phospholipidosis induction similar to HMA as demonstrated here and in our previous study [17]—as well as the decades-long clinical employment of amiloride and lack of non-specific HMA toxicity demonstrated in vivo and in vitro in these studies, it can be surmised that similar classes of LMP-inducing agents will be efficacious and well tolerated as clinical development expands. Along these lines, a recent evaluation of the CAD antihistamine clemastine reported LMP-mediated death of patient-derived glioblastoma cells but minimal cytotoxicity in normal human astrocytes [71], indicating a therapeutic window for treatment of this aggressive and lethal disease not unlike that observed in our HMA studies. Here, we have expanded on our previous observations that HMA selectively depletes transformed breast cancer cells by a lysosome-mediated mechanism, demonstrating HMA’s ablation of glioblastoma cells relative to non-transformed murine brain cells as well as significant drug-induced cytotoxicity across bladder, colorectal, liver, lung, pancreatic, and prostate cancer cells. These findings are significant considering the lack of therapeutic options and poor survival outcomes for patients diagnosed with these diseases, particularly glioblastoma and cancers of lung, liver, and pancreatic origin [58,59,60,61].

Beyond demonstrating the broad spectrum of heterogeneous cancers sensitive to HMA’s cytotoxicity, we established the translational relevance of our in vitro findings utilizing multiple ex vivo and in vivo models of breast cancer. We cultivated three-dimensional organoid systems to assess HMA’s effects on tumor tissue viability and further adopted a novel organotypic tumor slice culture technique to test drug response in tissues harvested from both genetically modified and syngeneic orthotopic xenograft mouse models of mammary tumorigenesis. Importantly, unlike other cell and tissue culture systems, tumor slice cultures preserve the morphology and heterogeneity of the original tissue as well as an intact tumor microenvironment for extended periods [51]. We also found that intratumoral HMA administration induced necrotic cell death in Met-1 mammary xenograft tumors—consistent with our in vitro findings [17]—and reduced metastasis to the lung, which prompted us to evaluate the impact of systemically delivered HMA/DCM nanoparticles in an experimental metastasis model. HMA/DCM inhibited metastatic lung colonization and outgrowth, which is significant as CSCs exhibit enhanced survival in the peripheral circulation and foreign tissue environments and are capable of differentiating and proliferating to form secondary tumors [74,75].

CSCs are critical drivers of tumor progression and metastasis and underlie key clinical challenges of recurrence and chemotherapy resistance, with increasing data demonstrating that CSCs can persist following targeted drug and immunotherapy administration. Critically, we found that HMA-induced LCD ablates heterogeneous tumor cell populations including chemotherapy insensitive ‘persister’ cells and breast, brain, bladder, colorectal, liver, lung, pancreatic, and prostate CSCs alike. BCSCs isolated based on cell surface marker expression or functionally enriched by tumorsphere cultivation from molecularly diverse breast cancer subtypes and mouse-derived primary tumor cells were uniformly susceptible to HMA. This broad cytotoxicity is highly attractive, as addressing issues of inter- and intratumoral heterogeneity in cancer therapy design remains an enormous challenge, and many CSC-targeting agents in development rely on tissue-specific marker expression [76,77]. Despite the pursuit of innovative solutions for CSC targeting including forced differentiation and CSC pathway inhibition, failure rates in clinical trials are high, with strategies largely limited by CSC plasticity and toxicity in normal stem cells [78]. Agents such as HMA and other CADs that subvert plasticity issues by effectively eliminating both total and CSC populations of heterogeneous tumor types offer tremendous potential in improving patient outcomes.

While few other studies have directly interrogated CSC cytotoxicity by lysosome-disrupting agents, a 2013 report showed co-expression of the lysosomal marker lysosome-associated membrane protein-1 (LAMP-1) with a marker of stemness in glioblastoma cells, indicating expansion of the lysosome compartment in CSCs [79]. Recently, the small molecule drug salinomycin was discovered to selectively kill CSCs through lysosomal iron sequestration, which catalyzed redox cycling and lysosomal ROS production, ultimately triggering LMP-mediated ferroptotic tumor cell death without displaying normal cell toxicity [80]. Together, these findings and our observations of HMA’s anti-CSC potency and transformation-selective effects support further investigation of lysosome-targeting drugs to meet the need for efficacious CSC-destroying agents. Potential applications of LMP inducers in clinical use are, in fact, wide-ranging, as CADs and other lysosome disrupting agents exhibit synergy with nanoparticles, proposed to enhance cellular internalization and targeting of existing anti-CSC compounds in development [78], by releasing them from sequestration in lysosomes [81]. Given transformation-associated changes to the lysosome compartment and mounting evidence that functional lysosomes contribute to multi-drug resistance, the lysosome is emerging as a compelling anticancer target.

## 5. Conclusions

When taken together with our previous observations, the current work demonstrates that HMA acts as efficiently toward therapy-resistant BCSC-like subpopulations as toward therapy-sensitive bulk breast cancer cell populations. These observations suggest that BCSC populations that are left intact by currently employed first-line chemotherapeutics may be sensitive to LMP-inducing agents. In this regard, CADs may be particularly useful as components of maintenance therapy regimens to eradicate residual CSCs following primary therapy, sidestepping synergistic toxicities of co-administered therapeutic agents. Moreover, cellular heterogeneity, plasticity, and related issues that confound the development of CSC-targeting agents drop out of consideration with agents that exert their cytotoxic effects based on transformation-dependent vulnerabilities rather than cellular subtype-specific properties. Indeed, our observations suggest that the LMP induction approach may be applicable to a variety of malignancies. While the modest potency and poor pharmacokinetic properties of HMA preclude its development as an antineoplastic, CADs or even amiloride derivatives that act via a similar mechanism but with greater potency offer promise in thwarting recurrence and metastatic spread.

## Figures and Tables

**Figure 1 cancers-14-00949-f001:**
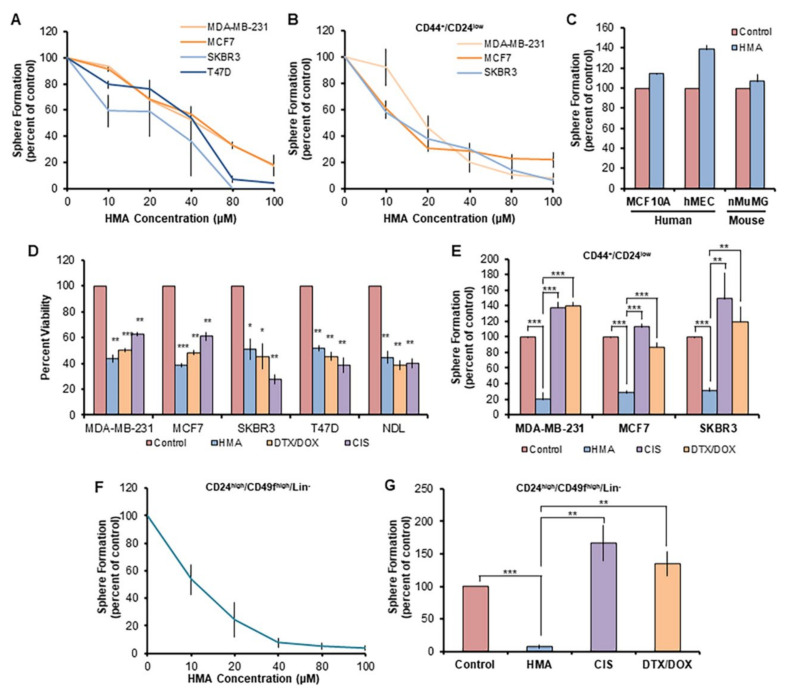
HMA ablates chemotherapy-resistant BCSCs but not normal mammary stem-like cells. (**A**) Secondary tumorsphere formation by human breast cancer cell lines was assessed after pretreatment with various concentrations of HMA for 24 h. Data are representative of three independent experiments. (**B**) Secondary sphere formation by CD44^+^/CD24^low^ BCSC-enriched human breast cancer cells after HMA pretreatment. (**C**) Secondary mammosphere formation of non-transformed human and mouse mammary epithelial cell lines was assessed after 40 μM HMA treatment for 24 h. (**D**) Total populations of MDA-MB-231, MCF7, SKBR3, and T47D cells were treated with vehicle (DMSO), 40 μM HMA, 40 μM cisplatin (CIS), or a combination of 170 nM doxorubicin (DOX) and 50 nM docetaxel (DTX) for 24 h, and cell viability was determined by trypan blue exclusion assay. (**E**) Sorted human BCSC cells were treated with vehicle, 40 μM HMA, 40 μM CIS, or 170 nM DOX with 50 nM DTX, and secondary sphere formation was determined. (**F**,**G**) Cells dissociated from MMTV-NDL mouse mammary tumors were sorted (CD24^high^/CD49f^high^/Lin^−^) to enrich for the BCSC population, and 7-day sphere formation following pretreatment with HMA (**F**) or standard chemotherapeutics (**G**) was quantified over four biological tumor replicates from independent mice. Error bars represent SEM. *, *p* < 0.05; **, *p* < 0.01; ***, *p* < 0.001, by Student’s *t*-test.

**Figure 2 cancers-14-00949-f002:**
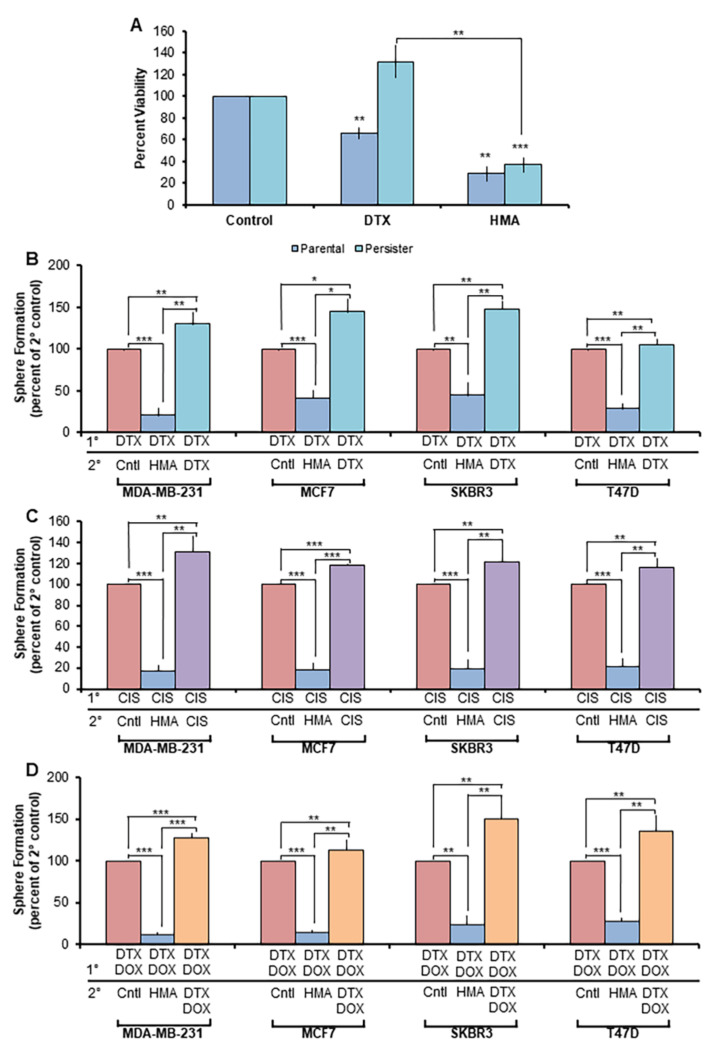
HMA is cytotoxic toward tumor cell populations insensitive to conventional chemotherapeutics. (**A**) Parental and DTX persister cells were treated with vehicle, 4 μM DTX, or 40 μM HMA for 24 h, and trypan blue exclusion assay was used to assess cell viability. Data are representative of five biological replicate experiments. (**B**–**D**) Breast cancer cell lines were treated with primary (1°) chemotherapy treatments—(**B**) 100 nM DTX, (**C**) 40 μM CIS, and (**D**) 170 nM DOX/50 nM DTX—for 72 h and then plated in serum-free, low-adherent conditions to enrich for the chemotherapy-insensitive BCSC population. Cells were subsequently treated with vehicle, 40 μM HMA, or a secondary (2°) chemotherapeutic treatment. Average sphere count was assessed after 7 days. Data are representative of three independent studies. Error bars represent SEM. *, *p* < 0.05; **, *p* < 0.01; ***, *p* < 0.001, by Student’s *t*-test.

**Figure 3 cancers-14-00949-f003:**
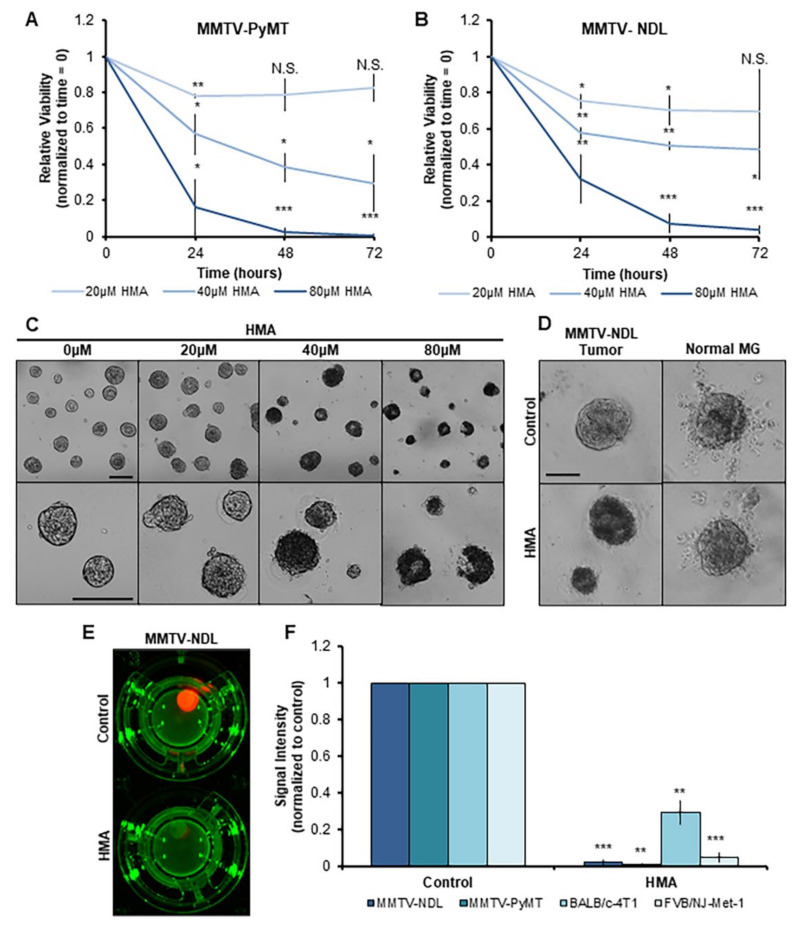
HMA impacts viability of mouse mammary tumor tissues ex vivo. (**A**–**D**) Organoids dissociated from MMTV-PyMT (**A**,**C**) or MMTV-NDL (**B**,**D**) mouse mammary tumors were administered increasing concentrations of HMA for 72 h. (**A**,**B**) RealTime Glo viability signal intensity was measured every 24 h and normalized to pretreatment organoid viability (time = 0) and vehicle control. Data encompass three biological tumor replicates from independent mice. (**C**) Representative images of MMTV-PyMT organoids following 72 h treatment are displayed. Scale bar = 200 μm. (**D**) Organoids derived from MMTV-NDL tumor (**left**) and normal mammary gland (**right**) were treated with 40 μM HMA or vehicle control for 48 h, and representative images are presented. Scale bar = 50 μm. (**E**,**F**) Tumor tissue slices from genetically modified mouse models of mammary tumorigenesis (MMTV-NDL, MMTV-PyMT) and orthotopic mammary xenograft models (BALB/cJ-4T1, FVB/NJ-Met-1) were exposed to vehicle (DMSO) and 40 μM HMA for 24 h, and overall tumor tissue viability was assessed by RealTime Glo exposure. (**E**) Representative images of luminescent viability signal measured in control- and HMA-treated MMTV-NDL tumor slices are shown. (**F**) Quantification of signal intensity indicating tissue viability over three to six tumor replicates from independent mice is shown. Signal was normalized to the intensity of each slice at time = 0, and then to vehicle control. Error bars represent SEM. *, *p* < 0.05; **, *p* < 0.01; ***, *p* < 0.001, by Student’s *t*-test.

**Figure 4 cancers-14-00949-f004:**
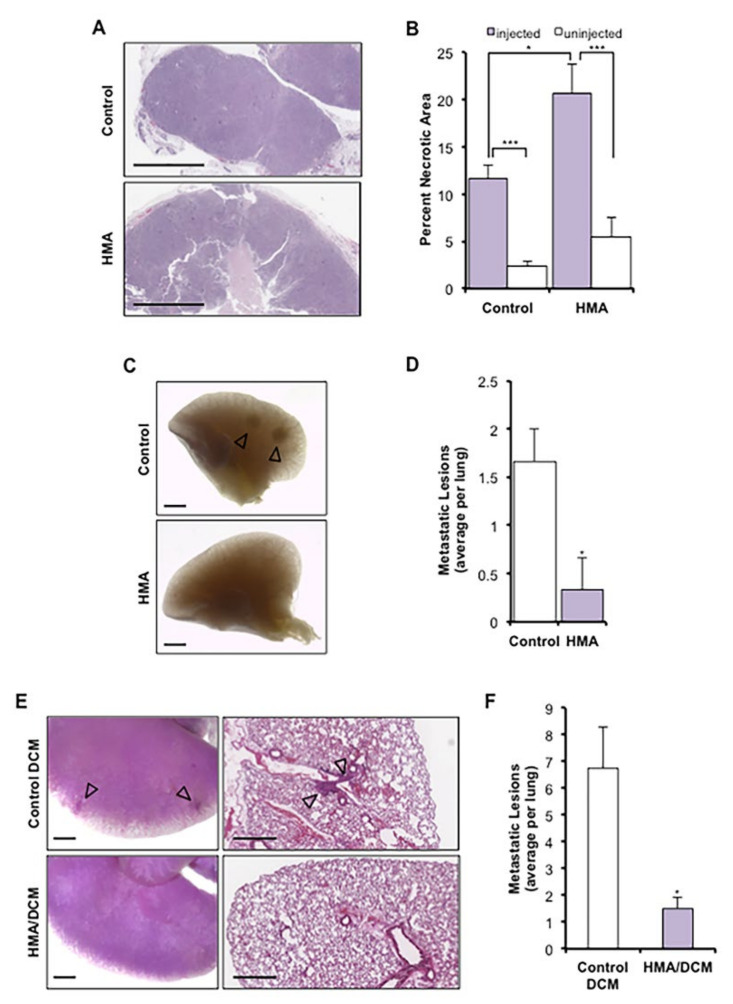
HMA induces necrosis in vivo and suppresses metastasis to the lung. (**A**–**D**) Met-1 xenograft mammary tumors received daily intratumoral injection of HMA (0.78 μg/mm^3^ tumor volume) or vehicle for 7 days (*n* = 3 FVB/NJ mice per cohort). (**A**) Representative H&E-stained sections; scale bar = 1 mm. (**B**) Necrotic area is presented as a percentage of total tumor area (mean ± SEM) and compared to un-injected contralateral (internal animal control) tumors. Significance was determined by Student’s *t*-test. (**C**) Carmine alum-stained lung tissue was evaluated for the occurrence of gross lesions (arrows); scale bar = 1 mm. (**D**) The numbers of gross metastatic lesions per mouse (mean ± SEM) are depicted for each treatment cohort, and significance was determined by Student’s *t*-test. (**E**,**F**) NDL tumor cells were instilled into the lateral tail vein of FVB/NJ mice, and animals were treated intravenously twice weekly for four weeks with HMA (30 mg/kg)-loaded DCMs (HMA loading: 5 mg/mL) or empty DCM (*n* = 4 mice per cohort). (**E**) Metastatic colonization of the lungs was determined by carmine alum staining; scale bar = 1 mm. (**F**) Quantification of metastatic lung lesions per animal (mean ± SEM). Significance was determined by Wilcoxon rank-sum test with continuity correction. *, *p* < 0.05; ***, *p* < 0.001, for all panels.

**Figure 5 cancers-14-00949-f005:**
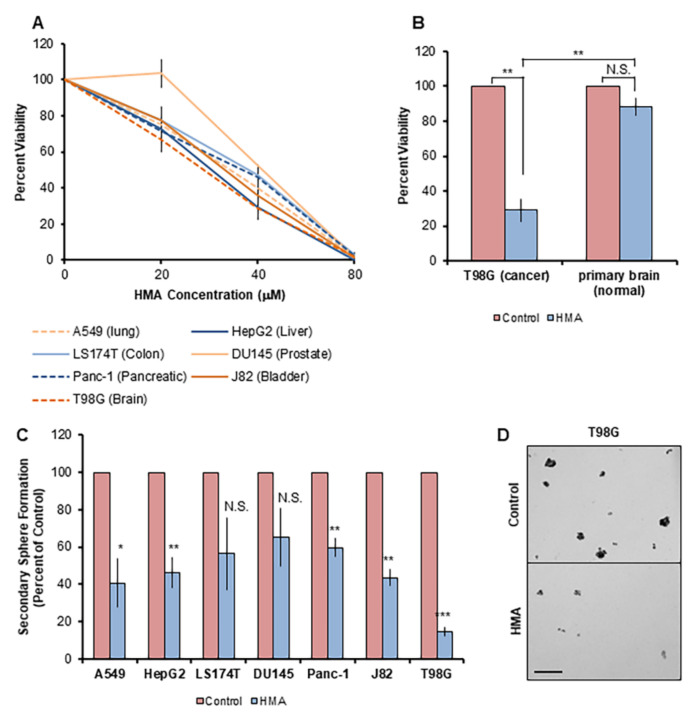
HMA depletes CSCs derived from an array of human tissue types. (**A**) The viability of human tumor cell lines treated with varying concentrations of HMA for 24 h was assessed by trypan blue exclusion assay. Data are presented as averages of at least three independent biological trials and expressed as a percent of vehicle control. (**B**) T98G glioblastoma cells and non-transformed mouse primary brain glial cells were subjected to 24 h treatment with 40 μM HMA. Cell viability representative of three replicate trials is normalized to vehicle control. (**C**) Cell lines were treated with vehicle or 40 μM HMA for 24 h and then subjected to the sphere formation assay. Secondary sphere formation is presented as the average sphere count of at least three independent biological experiments and normalized to vehicle control. (**D**) Representative images of DMSO control and 40 μM HMA-treated T98G spheres are displayed. Scale bar = 200 μm. Error bars represent SEM. *, *p* < 0.05; **, *p* < 0.01; ***, *p* < 0.001, by Student’s *t*-test.

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
