# Peer review of "The Cationic Amphiphilic Drug Hexamethylene Amiloride Eradicates Bulk Breast Cancer Cells and Therapy-Resistant Subpopulations with Similar Efficiencies"

_cancers, 2022, doi:10.3390/cancers14040949_

Round 1
Reviewer 1 Report
The manuscript by Berg et al, is well written with a sound experimental approach to describe the effect of hexamethylene amiloride in bulk breast cancer and breast CSC. The authors presented robust results to support their conclusions, such as the sphere-forming capacity. The work is very thorough and the results and interpretations are sound. Although the manuscript has merits to be published, these minor points must be addressed. Â
Â
- I wonder if the cells the second sphere formation are CSCs. The authors must show that the cells in the second sphere formation are CSCs.
- Authors have given voluminous data. Actually it is information overloaded. I do not understand the reason why authors have loaded the paper with voluminous data. Actually, figure 1 and 2, one gel bared and lose track the paper. It is information overloaded. Authors can try to give data in a manageable form.
- Numerous grammatical or typographic faults must be corrected.
Author Response
REVIEWER 1 – author responses in italics
The manuscript by Berg et al, is well written with a sound experimental approach to describe the effect of hexamethylene amiloride in bulk breast cancer and breast CSC. The authors presented robust results to support their conclusions, such as the sphere-forming capacity. The work is very thorough and the results and interpretations are sound. Although the manuscript has merits
to be published, these minor points must be addressed.
We thank the reviewer for their appreciation of our work.
• I wonder if the cells the second sphere formation are CSCs. The authors must show that the cells in the second sphere formation are CSCs.
Secondary sphere formation is the most rigorous in vitro assessment of stem-like properties of cells. Stem-like cells harbor the general property of survival under conditions of detachment, while secondary sphere formation provides evidence that the non-adherent population is capable of self-renewal [see refs 44-46].
• Authors have given voluminous data. Actually it is information overloaded. I do not understand the reason why authors have loaded the paper with voluminous data. Actually, figure 1 and 2, one gel bared and lose track the paper. It is information overloaded. Authors can try to give data in a manageable form.
The reviewer’s point is well-taken. While we feel that much of the presented data is necessary for rigorously demonstrating that drug impact on therapy-resistant cell populations is as efficient as on bulk populations, this focus is diluted toward the end of the Results section with the discussion of LMP. In alignment with Reviewer 2’s point that there are significant gaps in mechanistic understanding, we have opted to delete Figure 6 and its associated text to more sharply focus the manuscript on the drug’s cytotoxicity profile.
• Numerous grammatical or typographic faults must be corrected.
The manuscript has been checked for grammatical issues and spelling prior to resubmission
Reviewer 2 Report
The study of Berg and colleagues focused on in vivo, ex vivo and in vitro testing of anticancer effects of hexamethylene amiloride (HMA). The study builds on the previous knowledge on HMA anticancer effects performed by the group and provides an important contribution to the field. I find the study to be well conducted and described, and I would like to raise only a few minor points:Â
- In previous studies, authors studied HMA in the context of autophagy, and other groups also established conflicting evidence regarding HMA and type of cell death. I would recommend that authors address this either in the introduction or discussion. Particularly it would be interesting to explain if HMA effects are mediated via the increase of lysosomes or their permeabilization (or both). This remains unclear to the readers.Â
- Related to point (1), It would be of great interest to include a schematic representation of molecular pathways affected by HMA – illustrating the proposed mechanism of action and involved players. Particularly given to the fact that authors have already explored this topic in the past.
- In methodology sections, there are two animal sections used, it would be clearer to merge those sections and then highlight the difference in conducted in vivo study designs.Â
- Authors consider cells as resistant (called persister cells) after 9-day treatment with chosen agent. This seems to be arbitrarily chosen, and it is unclear if these cells are capable of division – did authors compare doubling times of persister cells with parent cell lines? Why would you consider this approach better than generating resistant cell lines via continuous exposure to the drug during multiple months?Â
- Please include growth factors used for MCF-10A cell line cultivation – this cell line has particular requirements for its growth (cholera toxin among others).
- Please include in the manuscript if the wellbeing of animals was monitored and if any signs of toxicity were observed.Â
- Why did you administer HMA intratumorally and how does this relate to the clinic? Usually, i.v., i.p. or oral routes of administration would be expected. How was the dose of HMA chosen/determined for in vivo study? Please include an explanation in the manuscript.
- Why does section 2.14. contain galectin translocation assay in italic?
Author Response
REVIEWER 2 – author responses in italics
Â
The study of Berg and colleagues focused on in vivo, ex vivo and in vitro testing of anticancer effects of hexamethylene amiloride (HMA). The study builds on the previous knowledge on HMA anticancer effects performed by the group and provides an important contribution to the field. I find the study to be well conducted and described, and I would like to raise only a few minor points:Â
We thank the reviewer for their appreciation of our work.
Â
- In previous studies, authors studied HMA in the context of autophagy, and other groups also established conflicting evidence regarding HMA and type of cell death. I would recommend that authors address this either in the introduction or discussion. Particularly it would be interesting to explain if HMA effects are mediated via the increase of lysosomes or their permeabilization (or both). This remains unclear to the readers.Â
We agree with the reviewer that the mechanism of action underlying HMA cytotoxicity is a key question. However, we also feel that this question deviates from the central thesis of the manuscript, that HMA as a cationic amphiphilic drug targets therapy-resistant and stem-like populations of breast cancer cells with the same efficiency as bulk cancer cells. Taken with Reviewer 1’s concern that the manuscript is information overloaded, we have removed the mechanistic Figure 6 in an effort to return the focus to drug cytotoxicity toward therapy-resistant cell populations.
Â
- Related to point (1), It would be of great interest to include a schematic representation of molecular pathways affected by HMA – illustrating the proposed mechanism of action and involved players. Particularly given to the fact that authors have already explored this topic in the past.
With the renewed focus on HMA cytotoxicity profile and away from LMP, we feel such a schematic may be better suited to a future manuscript more focused on mechanistic issues.
Â
- In methodology sections, there are two animal sections used, it would be clearer to merge those sections and then highlight the difference in conducted in vivo study designs.
These sections have been merged in the revised version of the manuscript.
Â
- Authors consider cells as resistant (called persister cells) after 9-day treatment with chosen agent. This seems to be arbitrarily chosen, and it is unclear if these cells are capable of division – did authors compare doubling times of persister cells with parent cell lines? Why would you consider this approach better than generating resistant cell lines via continuous exposure to the drug during multiple months?
The 9-day treatment paradigm was not arbitrarily chosen. ‘Persister cells,’ defined by shorter drug selection protocols such as 9 days, are a subpopulation of cells present within a wide range of tumor types that have been previously defined by other groups to survive cytotoxic exposure through reversible and non-mutational mechanisms.  Emerging evidence suggests that the persister cell pool constitutes a reservoir from which drug-resistant tumors may emerge (see refs 34,35). As with CSCs, their plasticity represents a significant barrier to efficient therapeutic targeting. The major conclusion of our study, that therapy-resistant and bulk populations are eradicated with similar efficiencies by CADs, suggests that the use of CADs may prove particularly effective toward CSC and persister subpopulations whose plasticity allows them to evade conventional treatments.
Â
- Please include growth factors used for MCF-10A cell line cultivation – this cell line has particular requirements for its growth (cholera toxin among others).
These details have been added to the Materials and Methods section 2.1.
Â
- Please include in the manuscript if the wellbeing of animals was monitored and if any signs of toxicity were observed.Â
We indicate in Supplementary Figure S3C that mouse body weights were stable over the course of treatment, and in the text of the revised manuscript that no signs of gross organ damage were observed, suggesting a lack of drug toxicity.
Â
- Why did you administer HMA intratumorally and how does this relate to the clinic? Usually, i.v., i.p. or oral routes of administration would be expected. How was the dose of HMA chosen/determined for in vivo study? Please include an explanation in the manuscript.
While the reviewer is correct in that intratumoral injection of HMA has no clinical relevance, our primary purpose with the experiment depicted in Figures 4A-D was to determine whether the necrotic cell death induced by HMA in vitro is recapitulated in animals. For the experimental metastasis model of Figures 4E and 4F, 30mg/kg was used based on the maximum tolerated dose of the naked drug in FvB strain mice. This is now indicated in Results section 3.4 of the revised manuscript.
Â
- Why does section 2.14. contain galectin translocation assay in italic?
This section has been deleted.
Â